# A structure-based extracellular matrix expansion mechanism of fibrous tissue growth

**Nicholas S Kalson\*, Yinhui Lu, Susan H Taylor, Tobias Starborg, David F Holmes, Karl E Kadler\***

Wellcome Trust Centre for Cell-Matrix Research, Faculty of Life Sciences, University of Manchester, Manchester, United Kingdom

**Abstract** Embryonic growth occurs predominately by an increase in cell number; little is known about growth mechanisms later in development when fibrous tissues account for the bulk of adult vertebrate mass. We present a model for fibrous tissue growth based on 3D-electron microscopy of mouse tendon. We show that the number of collagen fibrils increases during embryonic development and then remains constant during postnatal growth. Embryonic growth was explained predominately by increases in fibril number and length. Postnatal growth arose predominately from increases in fibril length and diameter. A helical crimp structure was established in embryogenesis, and persisted postnatally. The data support a model where the shape and size of tendon is determined by the number and position of embryonic fibroblasts. The collagen fibrils that these cells synthesise provide a template for postnatal growth by structure-based matrix expansion. The model has important implications for growth of other fibrous tissues and fibrosis.

**\*For correspondence:**
nickkalson@gmail.com (NSK);
karl.kadler@manchester.ac.uk
(KEK)

**Competing interests:** The authors declare that no competing interests exist.

**Reviewing editor**: Robb Krumlauf, Stowers Institute for Medical Research, United States

## Introduction

Tendons transmit tensile forces from muscles to bone and are amongst the heaviest loaded tissues in vertebrates; the forces have been estimated at ~ 16 kN/Kg of body mass (*Harrison et al., 2010*). The ability of tendons to transmit such large forces is directly attributable to an extracellular matrix (ECM) comprising collagen fibrils aligned parallel to the tissue long axis. Such forces occur during extended periods of activity with potentially devastating consequences on tissue integrity and cell survival. However, despite the high forces, tendon cells (tenocytes) remain viable and are able to sustain the integrity of the tissue. Our motivation for this study was to understand how tendons increase in size from an anlage of condensed mesenchyme to a mature tissue that is predominately ECM and which can withstand large forces without cell death.

The discovery of transcription factors expressed by tendon progenitors and the identification of signalling pathways in tendon has contributed greatly to our understanding of the earliest stages of tendon specification. However, the absence of a robust method of imaging cell-matrix organisation has precluded a detailed study of how the cell-rich anlage found in embryogenesis grows into a mechanically strong connective tissue in the adult. Advances in serial block face-scanning electron microscopy (SBF-SEM) provide a method to address this problem (*Starborg et al., 2013*). SBF-SEM is a practical addition to serial section reconstruction and can typically provide data at ~10 nm resolution in the sectioning plane for ~100,000 $\mu m^3$ volume of tissue, which is sufficient to quantify cell number, cell shape, and cell–cell interactions, as well as estimate the number and organisation of individual collagen fibrils. When used in combination with a developmental series, SBF-SEM provides new information on the changes in cell organization and matrix assembly during tissue growth.

**eLife digest** Young animals are able to grow in a way that allows them to maintain roughly the same shape until they reach their adult size. The growth of embryos is driven by increases in cell size and number, but it is less clear how the body grows after birth. By this point, many of the cells in the body are part of tendons and other fibrous tissues, where they are surrounded by a mesh of fibres made of collagen and other proteins. These fibres provide strength to the tissue, but may also restrict its ability to grow.

Tendons connect muscles to bones. They contain fibres of collagen that run along their length, which enables them to cope with very strong pulling forces. Kalson et al. used electron microscopy to generate highly detailed three-dimensional models of mouse tendons at three stages: in the embryo, at birth and six weeks later.

The experiments identified two stages in tendon development. During the first stage, the number of cells and fibres across the tendon is determined in the embryo. The fibres also slightly expand in diameter and form regular waves called crimps that are important for the structural strength of the tendon. The second stage happens after birth, during which the number of cells and fibres remains constant, but the tendons continue to grow because the fibres increase in diameter and length. The cells also move to form towers of cells running along the tendon.

From these observations, Kalson et al. propose that the numbers and locations of the cells and collagen fibres that determine the shape and size of tendons are established in the embryo. The collagen fibres create a framework for the continued growth of the tendon after birth. Future challenges are to understand how the number and the arrangement of cells in the tendon is determined before the collagen fibres are made, and how these cells control the number of collagen fibres that form.

The discovery of Scleraxis (Scx), amongst all other transcription factors, transformed studies of tendon development (*Brent et al., 2003*). The generation of Scx-GFP mice showed the precise location of tendons in the proximal–distal axis of the limb (*Schweitzer et al., 2001*). Scx specifies important interactions between somatic muscle and cartilage cell lineages leading to tendon development (*Brent et al., 2003*; *Sugimoto et al., 2013a*). However, Scx is also expressed in non-tendon tissues including ligaments, intervertebral discs, joints, and cartilage around the chondro-tendinous/ligamentous junction (*Sugimoto et al., 2013b*). Tenomodulin, a type II transmembrane glycoprotein, is expressed in tendons and ligaments and is a regulator of tenocyte proliferation and is involved in collagen fibril maturation (*Docheva et al., 2005*). Further studies have shown that Mohawk (*Liu et al., 2010*), EGR1, and EGR2 are transcription factors that are also involved in tendon formation (*Lejard et al., 2011*). Furthermore, the ability of EGR1 to promote tendon differentiation is partially mediated by TGFβ2 (*Guerquin et al., 2013*). Two main signalling pathways, TGFβ and FGF, have been identified in vertebrate tendon development (*Maeda et al., 2011*, and for review see *Tozer and Duprez, 2005*; *Schweitzer et al., 2010*). Collagen type I occurs in virtually all fibrous tissues along with type III, V (*Birk and Mayne, 1997*), XII, XIV (*Ansorge et al., 2009*), fibronectin and small proteoglycans such as decorin (*Berenson et al., 1996*; *Zhang et al., 2006*). Therefore, the expression of these transcription factors and structural proteins, and the presence of particular signalling pathways, does not, in itself, provide information on the development of tendon structure and function.

The defining event that signifies the onset of functional tendon development is the appearance of collagen fibrils in the ECM. The distribution of collagen fibril diameters distinguishes two stages in tendon development. In mouse, stage 1 begins at E12.5 when narrow (~35 nm) diameter collagen fibrils are formed within actin-dependent fibripositors at the cell surface (*Canty et al., 2004*, *2006*; *Kalson et al., 2013*). The length and width of the tissue doubles in a few days, and is accompanied by two-orders of magnitude increases in elastic modulus and ultimate tensile strength (*McBride et al., 1985*, *1988*). Serial section transmission electron microscopy (ss-TEM) of chick embryonic tendon showed that the tendon matrix contains bundles of collagen fibrils that undergo gradual rotation over several microns (*Birk and Trelstad, 1986*; *Birk et al., 1989*). Moreover, the collagen fibril bundles are stabilised by cell–cell connections containing cadherin-11 (*Richardson et al., 2007*). During stage 1

there is a modest increase in fibril diameter (to ~40 nm), which has been replicated in vitro by slow stretching of a 3-dimensional (3D) tendon-like construct containing embryonic tenocytes (*Kalson et al., 2011*). At birth (in the mouse), the unimodal distribution of narrow fibrils is quickly (within a few days) replaced by a bimodal distribution of fibril diameters with a range of 35–400 nm (*Goh et al., 2012*). The transition from unimodal to bimodal distributions specifies the onset of stage 2.

In addition to increased fibril diameter and changes to cell morphology, the collagen fibrils in tendon develop regular undulations, known as crimp (*Diamant et al., 1972*). Crimp becomes evident early in embryonic development (*Shah et al., 1982*) and provides the biomechanically important 'toe-region' to the force–extension curve (*Shah et al., 1982*). Crimp structure has been investigated by plane polarised light microscopy and by SEM (*Raspanti et al., 2005*; *Franchi et al., 2007*), but the structure of crimp is still debated; numerous studies describe a planar 2D undulating crimp, but mathematical modelling of crimped collagen fibrils in tendon suggests crimp to be helical in structure (*Grytz and Meschke, 2009*). Again, technical limitations of techniques used to study crimp have prevented a definitive description; for example, SEM requires chemical treatment of tissue to remove cellular material, precluding analysis of the composite 3D architecture.

Although ss-TEM provided insights into the short-range hierarchical structure of tendon, technical challenges of generating undistorted serial sections and achieving an absolute alignment have been major hurdles in studying hierarchical organisation beyond the level of a few cells. We here describe SBF-SEM studies of tendon development and show 3D reconstructions at three key stages of tendon development: stage 1 (embryonic), the stage 1–2 transition (newborn), and stage 2 (6 week). We interpret our findings as suggesting that the hierarchical structure of tendon, with collagen fibrils organised into fibril bundles with a biomechanically important spiral crimp structure, and the number of collagen fibrils in bundles, is established during embryogenesis. Subsequent postnatal tendon growth is achieved by increase in collagen fibril diameter and fibril length, likely by interface-limited molecular accretion, during which the spatial relationship between cells, fibril bundles and the crimp spiral, is maintained.

## Results

### Quantitation of collagen fibrils into bundles

To learn more about tendon development we examined embryonic day 15.5 (E15.5), newborn and 6 week postnatal mouse-tail tendon. First we investigated the collagen fibrils in the ECM, using TEM of ultra-thin sections cut perpendicularly to the tendon long axis, in which the collagen fibrils are transected transversely (*Starborg et al., 2013*). The results showed that at E15.5 the vast majority of collagen fibrils had near-circular outlines and were therefore perpendicular (to a close approximation) to the tendon long axis. The fibrils were organised in small groups, which occurred in channels defined by the plasma membranes of surrounding cells. These groups of fibrils are termed 'fibril bundles' (*Figure 1A*). At this stage of development the collagen fibrils are being synthesised in fibripositors at the cell-matrix interface (*Kalson et al., 2013*).

At birth the number of fibril profiles in fibril bundles increases >fourfold, and the fibril bundles are well defined by cellular plasma membrane extensions (*Figure 1B*). It is important to note that fibril profiles are fibrils in transverse section. It was not possible to count entire fibrils because some are too long to be contained within the SBF-SEM volumes we studied. TEM images showed electron dense cell–cell junctions between adjacent cell membrane extensions (*Waggett et al., 2006*; *Richardson et al., 2007*). In 6 weeks postnatal tendon, fibril bundles have grown in lateral size and are separated by striking elongated cell processes (*Figure 1C*). Analysis of TEM images demonstrated that fibril diameter increased during development, with mean fibril diameter increasing from 35.4 nm ± 0.2 nm at E15.5, to 46.5 ± 1.1 nm at birth and 160.1 ± 3.4 nm at 6 weeks postnatal (*Table 1*, *Figure 2A,B,C*). A bimodal distribution of diameters was found in 6 week postnatal tendon, as previously described (*Parry et al., 1978*). Notably, the increase in fibril diameter occurred uniformly throughout the ECM, and was not confined to fibrils close to the cell. This simple, key observation is critical to understanding how collagen fibrils grow in diameter independent of distance from the source of newly synthesized collagen (i.e., the cell).

Analysis of the bundles showed a pronounced increase in the number of fibril profiles per bundle during E15.5 to newborn, which is the period that we know de novo fibril assembly is occurring (*Table 1*) (*Trelstad and Hayashi, 1979*; *Kalson et al., 2013*). After birth the number of fibrils

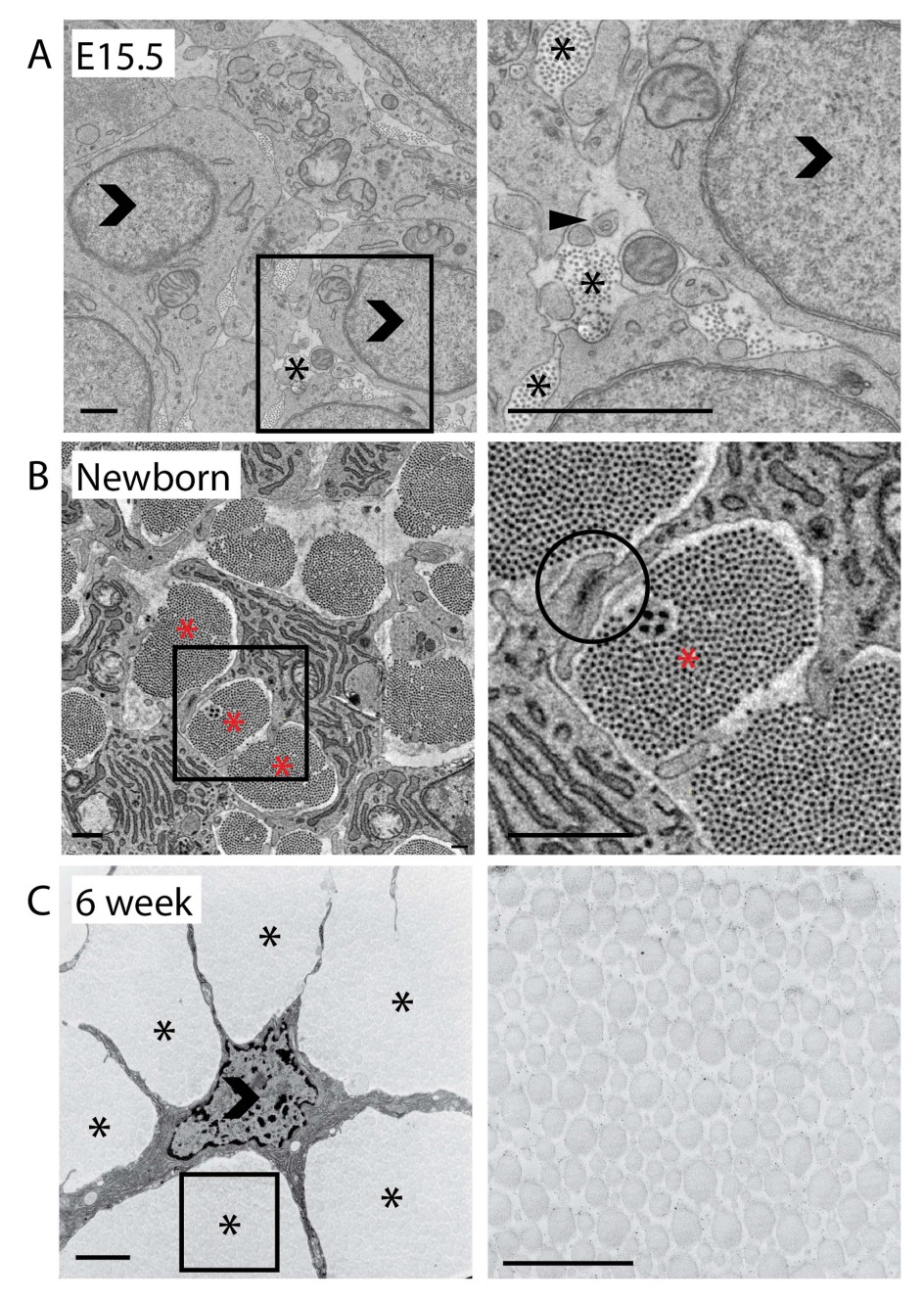

**Figure 1**. 2D analysis of mouse-tail tendon through development. (**A**) Transmission electron microscope (TEM) images of transverse sections at embryonic day 15.5 (E15.5). A closed arrowhead marks a fibripositor, seen during embryonic development but absent in postnatal tissue. Fibril bundles (marked *) are small. An open arrowhead marks a cell nucleus. (**B**) Newborn (P0). A cell–cell junction is circled between two cytoplasmic processes enclosing a fibril bundle. Bundles are larger and contain more fibrils than at E15.5. (**C**) 6 weeks postnatal. Cells have elongated processes that divide the matrix into bundles and connect with other cells. Bundles are larger than in newborn tissue. Open arrowhead marks cell nuclei. Stars (*) mark fibril bundles. Scale bar = 1 μm. The areas marked with a square in the left image in **A**, **B** and **C** are magnified in the right image.

stabilised; approximately 700 fibrils were found in each fibril bundle at birth and also at 6 weeks postnatal (*Figure 2D*, p = >0.1, Data in *Figure 2—source data 1*). At the same time as the number of fibril profiles per bundle had stabilised the number of fibril bundles per nucleus was also constant

**Table 1.** Summary of data

| Stage | Mean fibril diameter (nm) | Mean fibril area (nm²) | Mean number of fibrils per bundle | Mean fibril length (µm) | FAF (%) | Crimp length (µm) | Crimp helix radius (µm) | Tail length (mm) |
|---|---|---|---|---|---|---|---|---|
| E15.5 | 35.4 ± 0.2 | 1006.5 ± 13.8 | 156 ± 11 | 125 ± 18 | 1.4 ± 0.3 | – | – | 6.1 ± 0.30 |
| Newborn | 46.5 ± 1.1 | 2137.9 ± 370.9 | 659 ± 23 | 578 ± 17 | 17.6 ± 1.3 | 14.0 ± 0.3 | 1.6 ± 0.1 | 10.9 ± 0.9 |
| 6W | 160.1 ± 3.4 | 24,606.8 ± 933.0 | 684 ± 22 | 1250* ± 305 | 76.4 ± 1.3 | 99.3 ± 2.5 | 2.3 ± 0.1 | 67.6 ± 0.3 |

*For fibrils of diameter <150 nm.

Raw data are provided in **Supplementary file 1**.

(7.4 ± 0.3 bundles per cell vs 7.3 ± 0.3 in newborn vs 6 week, p = >0.1, *Figure 2E*). Fibripositors (sites of fibril formation during stage 1) were absent from 6 week tail tendon. The number of fibril profiles per µm² fell by a factor of ~10 during development as the diameter of the fibrils increased (*Figure 2F*) and fibril area fraction (FAF) increased during development from 1.4 ± 0.3% at E15.5, to 17.6 ± 1.3% at newborn, to 76.4 ± 1.3% at 6 weeks (*Figure 2G*). Corresponding with increase in fibril diameter, the mean fibril area increased from 1006.5 ± 13.8 nm² at E15.5, to 2137.9 ± 370.9 nm² at newborn to 24,606.8 ± 933.0 nm² at 6 weeks (*Figure 2H*).

## Change in cell shape during tendon development

TEM provides detailed ultra-structural information in two dimensions but does not easily permit study of 3D architecture, which is critical to the biomechanical function of collagenous connective tissues. We therefore used SBF-SEM to extend our studies of tendon development (*Starborg et al., 2013*). We studied mouse-tail tendon at the same developmental time points as *Figure 1*. Representative reconstructions of SBF-SEM datasets are presented in *Figure 3*. A transverse view (perpendicular to the long axis of the tendon) is shown in *Figure 3A*. The cells at E15.5 are rounded and close together. The cell bodies move apart as fibril bundles increase in size, eventually becoming star shaped (in transverse section) at 6 weeks postnatal. Longitudinal views (B and C) show the close relationship of cells, which are stacked on top of each other, thereby maintaining the precise longitudinal alignment of the fibril bundles. This end-to-end relationship of cells is maintained throughout development and is particularly striking in 6 week-old tendon. Reconstructing the fibril bundles in three dimensions (in red in *Figure 3C*) showed that in newborn and 6 week tendon the fibril bundles had a regular wavy configuration (crimp). The fibril bundles reconstructed in red (*Figure 3C*) are shown in the corresponding SBF-SEM images (*Figure 3D*).

## Increase in cell surface area during tendon development

The close relationship during embryogenesis between the cell membrane and fibril bundles led us to investigate 3D cell shape using SBF-SEM (*Figure 4*). SBF-SEM images at each of the time points studied are shown in the top row (*Figure 4A*—E15.5, *Figure 4B*—newborn and *Figure 4C*—6 week). One cell membrane is highlighted to demonstrate the membrane protrusions that form channels surrounding the fibril bundles. Longitudinal views demonstrate that the fibril bundle conforms to the shape of the cells; the cell and the fibril bundle remain in close apposition along the length of the cell. Contact between adjacent cell processes (which form the boundaries of these channels) is established during embryogenesis and is maintained during postnatal development of cell–cell contacts. The cell processes lengthen to accommodate the larger diameter collagen fibrils, resulting in a stellate (star shaped) appearance of the cell in transverse view in postnatal tendon (*Figure 4C*). Cell body length decreased during development from 60 ± 2 µm at E15.5, to 55 ± 3 µm at birth, to 27 ± 2 µm at 6 weeks (Data in *Figure 4—source data 1*). Calculation of cell dimensions demonstrated that cell volume remains relatively constant during this period of matrix growth between newborn and 6 weeks (*Figure 4D*), but the cell surface area increases greatly due to the growth of these long processes (*Figure 4E*). When isolated from tendon and plated on plastic the cell shape changed radically: in 2D cell culture the cells become flat and have the characteristic appearance of a fibroblast in culture (*Figure 4—figure supplement 1*). There was no difference in the surface area of the cells when cultured on plastic comparing E15.5 cells and 6 week cells (Data in *Figure 4—source data 1*).

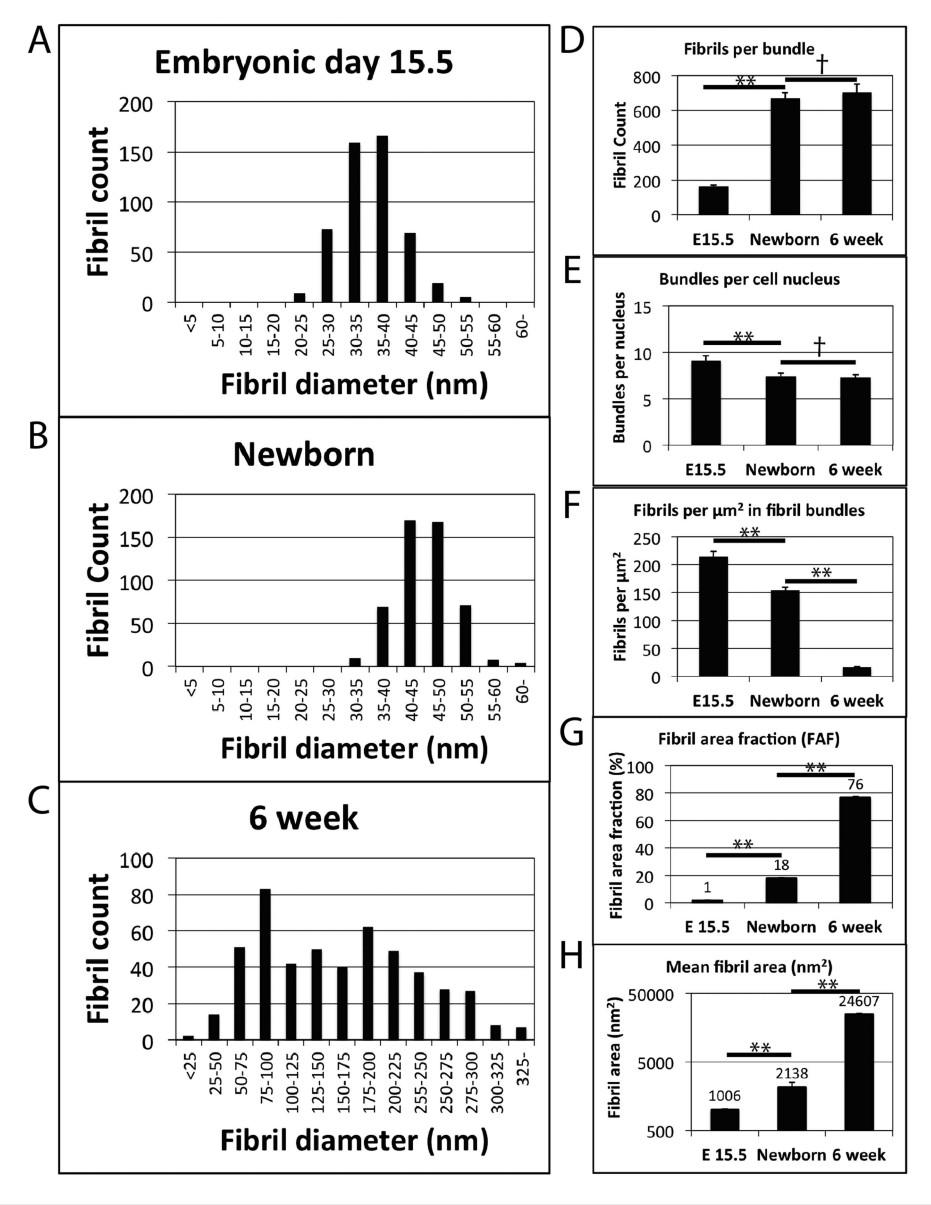

**Figure 2**. Fibril diameter and fibril bundling during development. (**A**) Fibril diameter plot for embryonic day 15.5 (**A**), newborn (**B**) and 6 week old tendon (**C**). In embryonic development there is a gradual increase in fibril diameter, which has a unimodal size distribution. Postnatal tendon has larger fibrils with a bimodal distribution. (**D**) There is a major (~fourfold) increase in fibril profiles in transverse section per bundle from E15.5 to newborn tissue. This fibril number remains constant postnatal in mature tissue. (**E**) Fibril bundle number per cell nucleus is constant postnatal. (**F**) The number of fibrils per $\mu m^2$ decreases as a function of increasing fibril diameter. (**G**) The area occupied by fibrils (as a proportion of total tendon area) increases significantly at each time point. (**H**) Mean fibril area increases significantly at each time point (derived from fibril diameter data assuming circularity). A logarithmic scale has been used on the y-axis. ** Indicates significant difference ($p = <0.05$), † indicates $p = >0.05$. Source data in *Figure 2—source data 1*.

The following source data is available for figure 2:

**Source data 1**. Fibril diameter and bundle data.

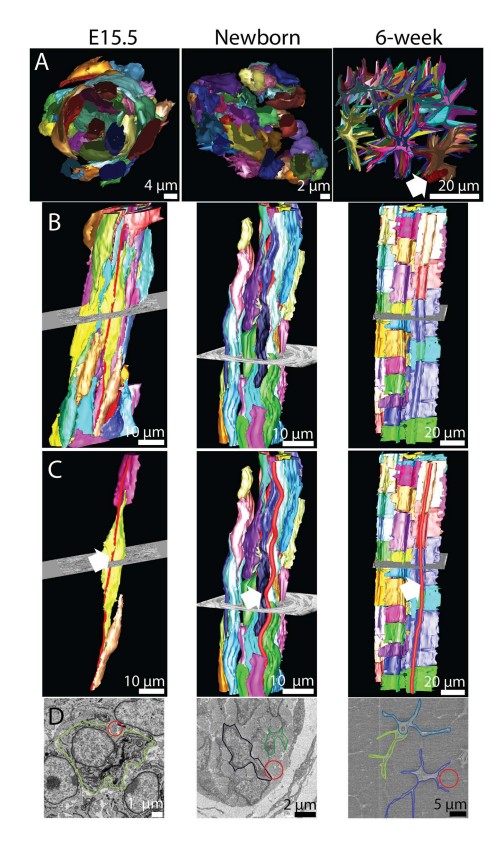

**Figure 3**. The 3D morphology of tendon changes from embryonic to postnatal tissue. 3D reconstructions from SBF-SEM images of E15.5, newborn and 6 week mouse-tail tendon. Individual cells are reconstructed in a different colour. (**A**) A transverse view of reconstructed tendon cells. E15.5 is comprised predominantly of rounded cells, with relatively little collagenous ECM. During development the ECM grows rapidly; by 6 weeks postnatal there are large spaces between cells containing bundles of collagen fibrils. One bundle (red) is marked with a white arrowhead. (**B**) Longitudinal views demonstrate the development of (1) well-defined channels formed by cells stacked end-to-end, and (2) wavy orientation of tenocytes corresponding to crimped bundles of collagen fibrils, particularly notable in newborn tendon. (**C**) The same reconstructions as in **B** with a bundle of collagen fibrils highlighted (red, marked with a white arrow). Longitudinally continuous bundles of fibrils are seen at E15.5 closely associated with tendon cell membranes. Crimp is clearly seen in newborn tendon and persists in 6 week tendon with a longer wavelength. (**D**) SBF-SEM images corresponding to **C**. The fibril bundle in **C** is circled red. Cell membranes are also outlined.

## Reduction in cell number per unit of tissue volume during development

To obtain a better understanding of the cell changes that underpin tendon growth we used SBF-SEM to analyse the change in cell number during embryonic and postnatal development (*Figure 5*, data in *Figure 5—source data 1*). We identified cell nuclei in 3D reconstructions from SBF-SEM data (*Figure 5A,B*). The number of cells per unit volume of tendon tissue fell significantly from $1.26 \pm 0.02$ to $0.91 \pm 0.02$ cells per 1000 $\mu m^3$ between E15.5 and birth, to $0.14 \pm 0.01$ at 6 weeks ($p = <0.05$, *Figure 5C*). Measurement of cells per unit area of transverse tendon area showed a significant fall from $30.1 \pm 0.6$ cells per 1000 $\mu m^2$ (E.15.5) to $20.5 \pm 0.8$ (newborn) to $1.9 \pm 0.1$ (6 week, $p = <0.05$, *Figure 5D*).

Tenocytes in newborn and 6 week tendon are arranged in stacks, between which sit bundles of collagen fibrils. The observation that cell body length decreases during development led us to measure the distance between cell nuclei in stacks. At birth cell nuclei were spaced $39.9 \pm 1.8$ $\mu m$ apart (on the longitudinal axis) and at 6-weeks they were $27.5 \pm 1.1$ $\mu m$ apart ($p = <0.05$, *Figure 5E*). The discrepancy between cell length and distance between nuclei in newborn tissue was due to overlap of cells in stacks (*Figure 5A*). These data suggest that there is a relative increase in the number of cells in longitudinal axis in cell channels during tendon growth comparing newborn with 6 week tendon. We also analysed the change in distance between adjacent cells in different cell stacks in transverse sections. The horizontal distance between cell nuclei increased from $6.2 \pm 0.2$ to $31.5 \pm 0.9$ $\mu m$ from birth to 6 weeks ($p = <0.05$, *Figure 5F*).

## Cell–cell contacts are maintained during development

The maintenance of similar numbers of fibril bundles per cell (in transverse section) during postnatal development and the formation of channels that defined fibril bundles by cell processes led us to investigate cell–cell contacts in more detail. At E15.5 no distinct cell membrane channels were formed via membrane protrusions contacting adjacent cells; instead groups of narrow-diameter fibrils were seen between adjacent cell membranes or enclosed in cell membrane invaginations (as shown in *Figures 1A, 6A* and *Figure 6—figure supplement 1A*, data in *Figure 6—source data 1*). Newborn tenocytes formed protrusions that contacted adjacent cells; these formed channels surrounding fibril bundles (*Figure 6B* and *Figure 6—figure supplement 1B*). Modeling of these cell–cell contact regions in 3D revealed

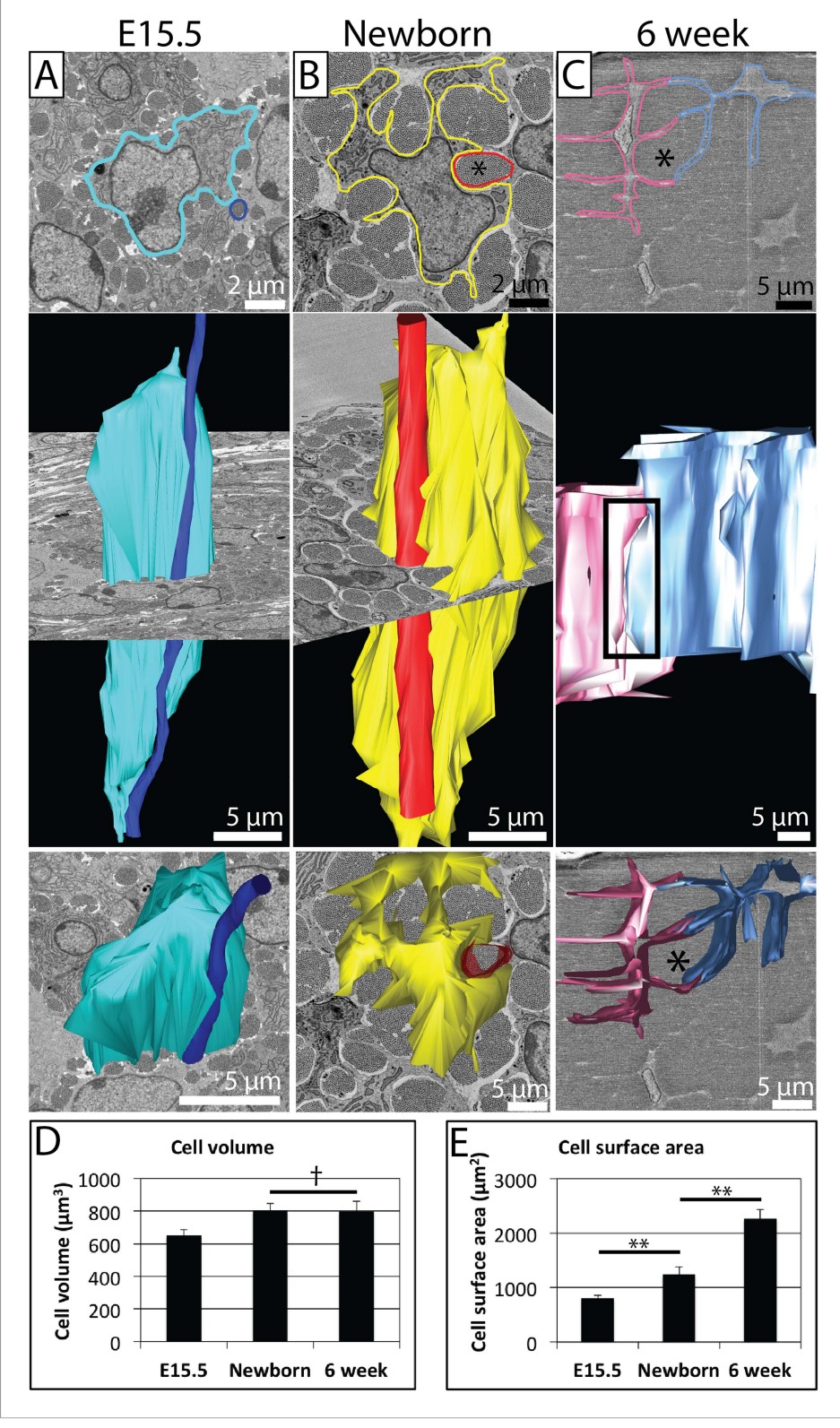

**Figure 4**. Changes in cell morphology during development. (**A**) E15.5 mouse-tail tendon. A cell membrane is highlighted in blue (top panel, single SBF-SEM transverse image) and reconstructed (middle and bottom panels). A fibril bundle, circled in dark blue (top), is reconstructed, and lies closely associated with the cell along the entire cell length. (**B**) Newborn mouse-tail tendon. A cell membrane is highlighted in yellow (top panel, single SBF-SEM

*Figure 4. continued on next page*

*Figure 4. Continued*

transverse image). A fibril bundle is highlighted in red and reconstructed in red, together with the corresponding cell (in yellow, middle and bottom panels). The fibril bundle is again closely associated with the cell. Cell protrusions surrounding fibrils are better defined than at E15.5. (**C**) 6 week old mouse-tail tendon. Two adjacent cell membranes are highlighted blue and pink (top panel, single SBF-SEM transverse section image). The same cells are reconstructed below. A fibril bundle is marked with *. The two cells have elongated processes that reach out and contact each other (highlighted with a rectangle, top reconstruction), enveloping a bundle of collagenous ECM (*). (**D**) There is a small increase in cell volume between E15.5 and newborn time points, but no further increase in cell volume, which remains relatively constant during postnatal development. (**E**) Cell surface area increases markedly (1.5 fold and 1.8 fold for E15.5 to newborn and newborn to 6 week old, respectively) as cells develop large cell membrane processes to enlarge the channels containing the fibril bundles. ** Indicates significant difference (p = <0.05), † indicates p = >0.05. * Marks a fibril bundle. Source data in *Figure 4—source data 1*.

The following source data and figure supplements are available for figure 4:

**Source data 1**. Cell morphology data.

**Figure supplement 1**. Cell shape in 2D culture.

close apposition of membranes along the longitudinal axis of the tissue, thereby defining the fibril channels. At 6 weeks a similar morphology of cell–cell contact was found, although the cell processes were much longer (*Figure 6C* and *Figure 6—figure supplement 1C*). Furthermore, immunofluorescence staining against connexins 32 and 43 revealed positive immunolocalisation of cell–cell junctions in mature (6 week) tendon (*Figure 6D,E*).

Quantitation of cell–cell contacts showed that in newborn tendon each tendon cell forms distinct channel-defining protrusions that contact, on average, 4.6 ± 0.1 adjacent cells. At 6 weeks each tendon cell contacts 4.5 ± 0.1 adjacent cells (p = >0.05, *Figure 6—figure supplement 1D*). The number of protrusions per cell that contacted adjacent cells was also similar in newborn and 6 week tendon (8.2 ± 0.2 vs 8.3 ± 0.2, *Figure 6—figure supplement 1E*).

## Fibril bundles have spiral crimp organisation

The observation that the fibril bundles had a regular undulating configuration (in *Figure 3*) led us to investigate further the 3D shape of the fibril bundles using SBF-SEM (*Figure 7*). Single transverse images taken with SBF-SEM are shown in *Figure 7A*. Fibril bundles are highlighted. These fibril bundles were reconstructed and shown in longitudinal views in *Figure 7B*. At E15.5 an undulation or wavy pattern is seen, but is neither regular nor well defined. By birth the fibril bundles have developed regular undulations, of defined and consistent wavelength. Importantly the undulations are in axial-register (see *Figure 7B*, middle panel). This wavy pattern is seen in 6 week tendon (*Figure 7B*, right panel). Oblique views of the reconstructed fibril bundles are shown in *Figure 7C* to demonstrate the spiral nature of the crimp structure, which is not clearly defined at E15.5, but is marked in newborn and 6 week old tendon. These data show the crimp to derive from a helical spring sub-structure, and is not simply a planar (2D) undulation. The wavelength of the crimp increases during development from 14 ± 0.3 µm at birth to 99 ± 2.5 µm at 6 weeks (*Table 1*, *Figure 7—figure supplement 1*, Data in *Figure 7—source data 1*). The crimp helix radius (the distance from the central axis) was 1.6 ± 0.1 µm at birth and 2.3 ± 0.1 at 6 weeks. The irregular pattern of the crimp at E15.5 precluded accurate measurement of crimp wavelength and translation distance.

To give an indication of the macroscopic degree of tissue growth we measured tail length (*Table 1*, Data in *Figure 2—source data 1*). Tail length increased from 6.1 ± 0.3 mm (E15.5) to 10.9 ± 0.9 mm (newborn) to 67.6 ± 0.3 mm (6 week); overall a factor of 10 increase. The increase in length between newborn and 6 weeks closely matches the change in crimp wavelength (~factor of 6–7 fold increase).

In 119 out of 120 specimens we have so far examined we have consistently found the helical coil to turn in a left-handed spiral. In one specimen from 6 week mouse we observed a right-handed helix. This continues the alternating handedness from polypeptide chains (left-handed helix), to collagen molecules (right-handed), to microfibrils (left-handed) and collagen fibrils (right-handed, as described in [*Holmes et al., 2001*]). These data show that the fibrils in tendon bundles then form a left-handed

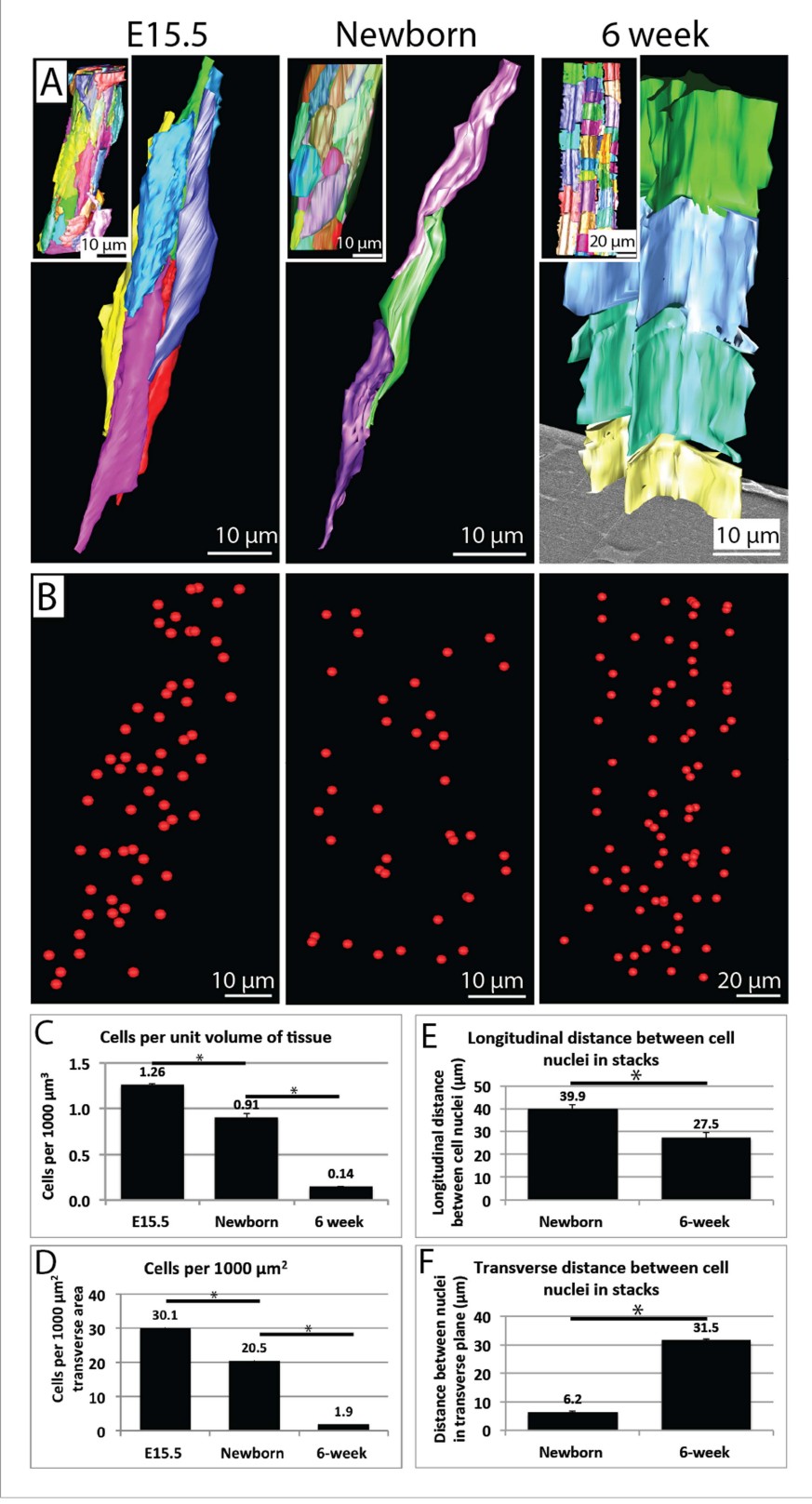

**Figure 5**. Cell number during development. (**A**) 3D reconstructions of longitudinal views of tendon tissue. Individual cell membranes are reconstructed in different colours. Inset images show a lower magnification view. Note the longitudinal-axis cell overlap in newborn tendon, which is not seen at 6 weeks. (**B**) Cell nuclei from reconstructions in *Figure 5. continued on next page*

*Figure 5. Continued*

(**A**) are marked with red spheres. (**C**) Calculation of cell number per unit volume of tissue. There is a significant decrease in cells per unit volume from E15.5 to newborn (1.26 ± 0.02 to 0.91 ± 0.02 cells per 1000 $\mu m^3$) and from newborn to 6 weeks (an almost sevenfold change, 0.91 ± 0.02 to 0.14 ± 0.01 cells per 1000 $\mu m^3$). (**D**) Calculation of cells per 1000 $\mu m^2$ of transverse tissue area. The number of nuclei in transverse section was determined. There was a significant decrease in cells per 1000 $\mu m^2$ from E15.5 to newborn and from newborn to 6 weeks. (**E**) The longitudinal distance between nuclei of cells in stacks decreases from newborn to 6 weeks (E15.5 cells are not stacked, precluding this analysis). (**F**) The transverse distance between cell nuclei in adjacent cell stacks increases significantly between newborn and 6 weeks. * Indicates p = <0.05. Source data in *Figure 5—source data 1*.
The following source data is available for figure 5:
**Source data 1**. Cell number data.

helical macrostructure. We have observed this left-handed spiral structure of fibril bundles in all the anatomically distinct tendons we have examined using SBF-SEM, including mouse Achilles tendon and chick metatarsal tendons, with different biomechanical properties compared with mouse-tail tendon, an axial tendon (*Benjamin et al., 2008*).

## Fibril length increases during development

Using SBF-SEM we were also able to make estimates of mean fibril length by tracking fibrils and counting fibril tips (*Starborg et al., 2013*). This demonstrated a significant increase in fibril length through development, from 125 ± 18 µm at E15.5, to 578 ± 17 µm in newborn tendon (*Figure 7—figure supplement 2*, p = <0.05, Data in *Figure 7—source data 1*). At E15.5 160 fibril tips were identified (within 10,000 µm fibril lengths tracked). In newborn mouse tendon, 69 tips were found (over ~20,000 µm fibril lengths tracked). At 6 weeks fibril tips were rare; 16 tips were identified within 10,000 µm fibril lengths tracked, all occurring in fibrils less than 150 nm in diameter (the smaller population of postnatal fibrils). No tips were found in fibrils >150 nm in diameter. This gave an estimate of mean-length of 1250 ± 305 µm for fibrils less than 150 nm diameter. We are unable to estimate the length of larger-diameter fibrils in 6 week tendon. These data, together with the information in *Figure 2* and *Table 1*, are interpreted in a model of fibril growth in *Figure 8*.

## Discussion

The events during tendon development when ECM overtakes cells as the major component of the tissue are poorly understood, largely because of the lack of a suitable technique with which to examine the micro- and macro-scale architecture. At a micro-scale, it was unknown how collagen fibrils increase in diameter from ~35 nm to ~400 nm (stage 1 to 2 transition), and curiously, why diameters are independent of the distance from the cell. At a macro-scale it was unknown how tenocytes become aligned in tramlines parallel to the tissue long axis in adult tendon, and how crimp develops. In this study we have shown that structure-based matrix expansion in the presence of stable cell-to-cell junctions is a novel mechanism for driving tissue growth.

The appearance of collagen fibrils and thus the start of tendon development proper, coincides with the formation of extracellular channels whose borders are delineated by cellular extensions (*Birk and Trelstad, 1985*). As shown in *Figure 1*, the wrapping of plasma membrane extensions around bundles of collagen fibrils in newborn and 6 week tendon produces a highly convoluted cross-sectional shape of the cell. Contacts between projections of neighbouring cells are stabilised by cell junctions (*McNeilly et al., 1996*; *Ralphs et al., 2002*; *Richardson et al., 2007*). Thus, neighbouring cells define vertical channels into which newly formed collagen fibrils are assembled, thereby generating bundles of collagen fibrils that are corralled into lying parallel to the tendon long axis.

Although there is an increase in the number of cells longitudinally (in cell stacks) between newborn and 6 weeks, the fall in cells per unit volume of tissue due to expansion of the ECM suggest that the major driver of tissue growth is an increase in the ECM in channels between cell stacks. The extended longitudinal cell–cell contacts seen in both newborn and in 6 week tendon tissue, and the maintenance of similar numbers of cell–cell contacts and cell protrusions at birth and 6 weeks, suggests that the cell–cell contacts formed in embryogenesis form the basic pattern for the mature

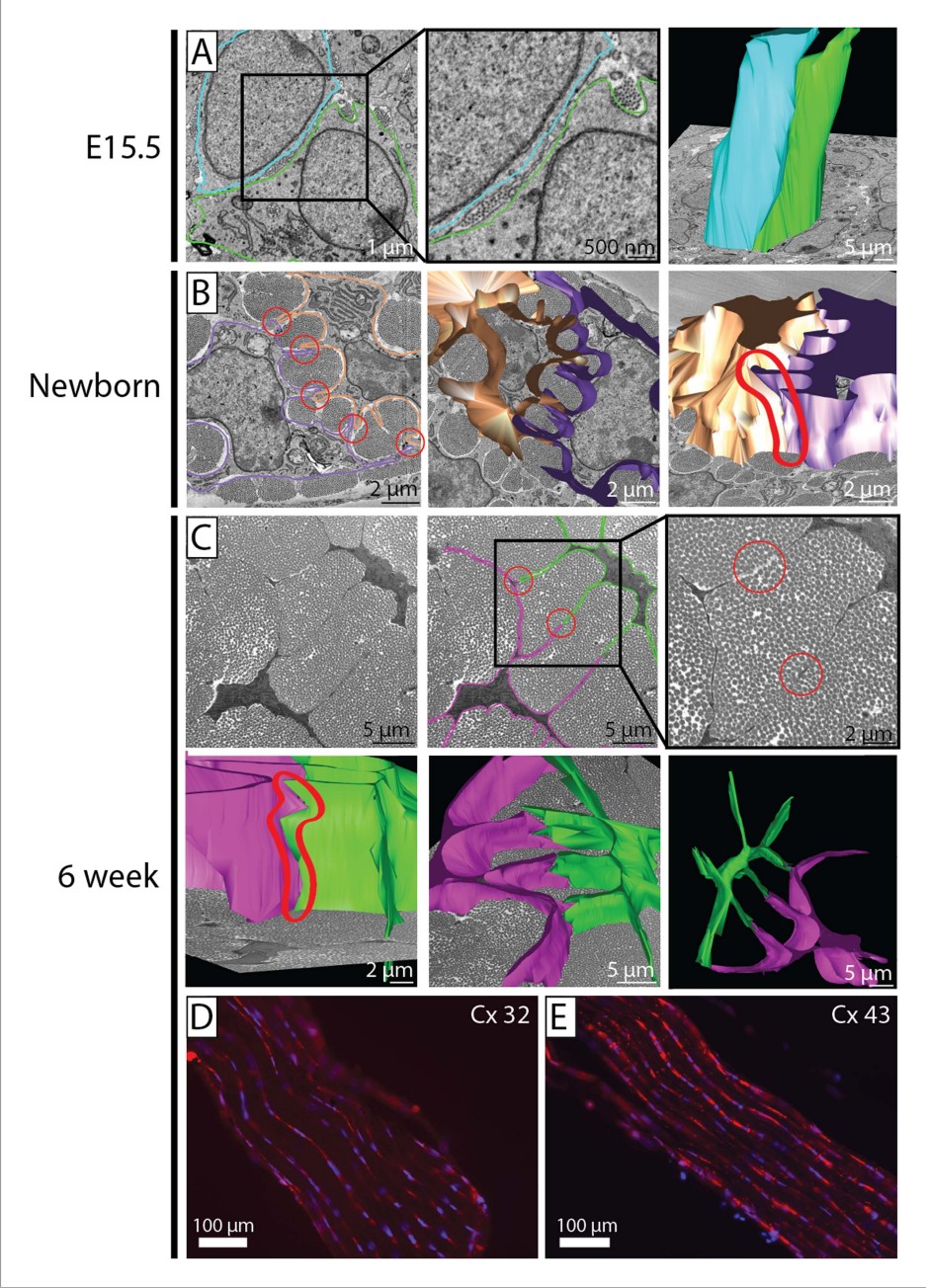

**Figure 6**. Cell–cell contacts during development. (**A**) SBF-SEM images of E15.5 (left and middle panel), 3D cell reconstruction (right panel). Two cells are outlined (blue and green tracing their cell membranes). At E15.5 cells are packed closely, without distinct membrane protrusions forming well-defined channels for collagen fibrils. Narrow diameter fibrils are found between adjacent cell membranes and in cell membrane invaginations. 3D reconstruction of E15.5 cell membranes shows the cells to be in close apposition along their entire length. (**B**) SBF-SEM images of newborn tendon cells. Two cell membranes are traced (brown and purple). Cell-protrusions are seen, forming channels for ECM fibrils. (**C**) SBF-SEM images of 6 week tendon. Two cell membranes are traced (pink and green, upper row, three images), and reconstructed (lower row, three images). Distinct cell–cell contacts are seen between adjacent cells (circled in red in SBF-SEM images and in the 3D reconstruction). (**D**, **E**) Positive immunofluorescence staining for connexin 32 and connexin 43 in 6 week tendon tissue.

The following source data and figure supplements are available for figure 6:

*Figure 6. continued on next page*

*Figure 6. Continued*

**Source data 1**. Cell-contact data.

**Figure supplement 1**. Quantification of cell–cell contacts.

tendon tissue. Positive immunolocalisation of important gap junction components, connexin 32 and 43, at 6 weeks supports this conclusion.

The increase in cell number along the longitudinal axis of the tendon in postnatal growth is compatible with this interpretation. New cells formed by mitosis during tissue growth can be added in cell stacks whilst at the same time maintaining the longitudinal channels containing fibril bundles, which are growing laterally due to a dramatic increase in fibril diameter.

We were able to quantitate the number of fibril profiles, diameter and average length of collagen fibrils during embryonic to postnatal development. As shown in *Figure 2*, fibril diameters increased during E15.5 to newborn but maintained a unimodal diameter distribution. During embryonic development the number of fibrils per bundle increases (from E15.5 to birth). There was a significant increase in fibril diameter in postnatal tendon, and fibrils developed a bimodal diameter distribution. Also during postnatal growth (between birth and 6 weeks) the number of bundles per cell remained constant and the number of fibril profiles per bundle remained constant. The fact that the number of bundles per cell remained constant was a good indication that the cell–cell junctions between neighbouring cells are stable. SBF-SEM allowed estimation of fibril length, which showed a significant increase in fibril length from E15.5 to newborn. Taken together, these data showed that growth during the period E15.5 to newborn was the result of new fibril formation together with an increase in fibril diameter and length. Recent studies have shown that the collagen fibrils in the extracellular channels, at this early stage of tendon development, are formed in cell-surface fibripositors (*Canty et al., 2004*; *Kalson et al., 2013*), and therefore, the cell strictly controls the number of fibrils deposited to the bundles.

SBF-SEM analysis of 6 week postnatal tendon provided explanations for features that have long been a puzzle. As shown in *Figure 2*, the number of bundles per cell and the number of collagen fibril profiles per bundle remain stable between birth and 6 weeks postnatal. During the same period, fibril diameters increase markedly and the distribution becomes bimodal. These data showed that, in mouse, cells set the number of bundles and number of collagen fibrils per bundle at the time of birth. Local growth in tendon size between newborn and 6 weeks postnatal is driven primarily by increase in collagen fibril diameter and length. Growth models have been proposed for the increase in fibril diameter based on inter-fibrillar fusion and accretion of newly-synthesised collagen molecules (*Birk et al., 1995*; *Kadler et al., 2000*; *Trotter et al., 2000*). Inter-fibrillar fusion can potentially involve tip-to-tip, tip-to-shaft, and shaft-to-shaft fusion. Tip-to-tip fusion results in fibril lengthening. Quantitative mass mapping by scanning transmission electron microscopy and analysis of fibril staining patterns by TEM showed that tip-to-tip fusion occurs during early embryonic tendon morphogenesis and relies on unipolar collagen fibrils, in a process that is regulated by collagen-proteoglycan interactions (*Graham et al., 2000*). Tip-to-shaft fusion also occurs, which generates branched networks (*Kadler et al., 2000*). A model for linear and lateral growth of fibrils during tendon development has been proposed in which decorin, and perhaps related small proteoglycans, regulates surface interactions of participating fibrils (*Birk et al., 1995*). Thus, fibril fusion is regulated by the directionality of collagen molecules in the fibril (*Kadler et al., 1990*), the availability of tips from unipolar fibrils (*Graham et al., 2000*), and by the presence of proteoglycans at the fibril surface (*Birk et al., 1995*; *Graham et al., 2000*). An important and often overlooked consequence of fibril fusion is the inevitable reduction in fibril numbers as fibrils fuse together, and the decrease in the number of fibrils per cell. It is also to be noted that fibril fusion has no effect on fibril volume fraction as no new fibril volume is created by any type of fusion; we found very few fibril fusion (tip-shaft) events when we tracked fibrils with SBF-SEM. The data presented here show that collagen fibril numbers (in profile) increased during stage 1 of development and remained constant during stage 2, despite increases in fibril diameter (in stage 1 and stage 2). Although we cannot rule out tip-to-tip fusion (which would not be expected to affect fibril diameter) our diameter and fibril number data do not support shaft-to-shaft fusion as a major contributor to increases in fibril diameter during tendon development. We also have not observed irregularly shaped fibrils, which would be expected to occur if lateral fusion was occurring.

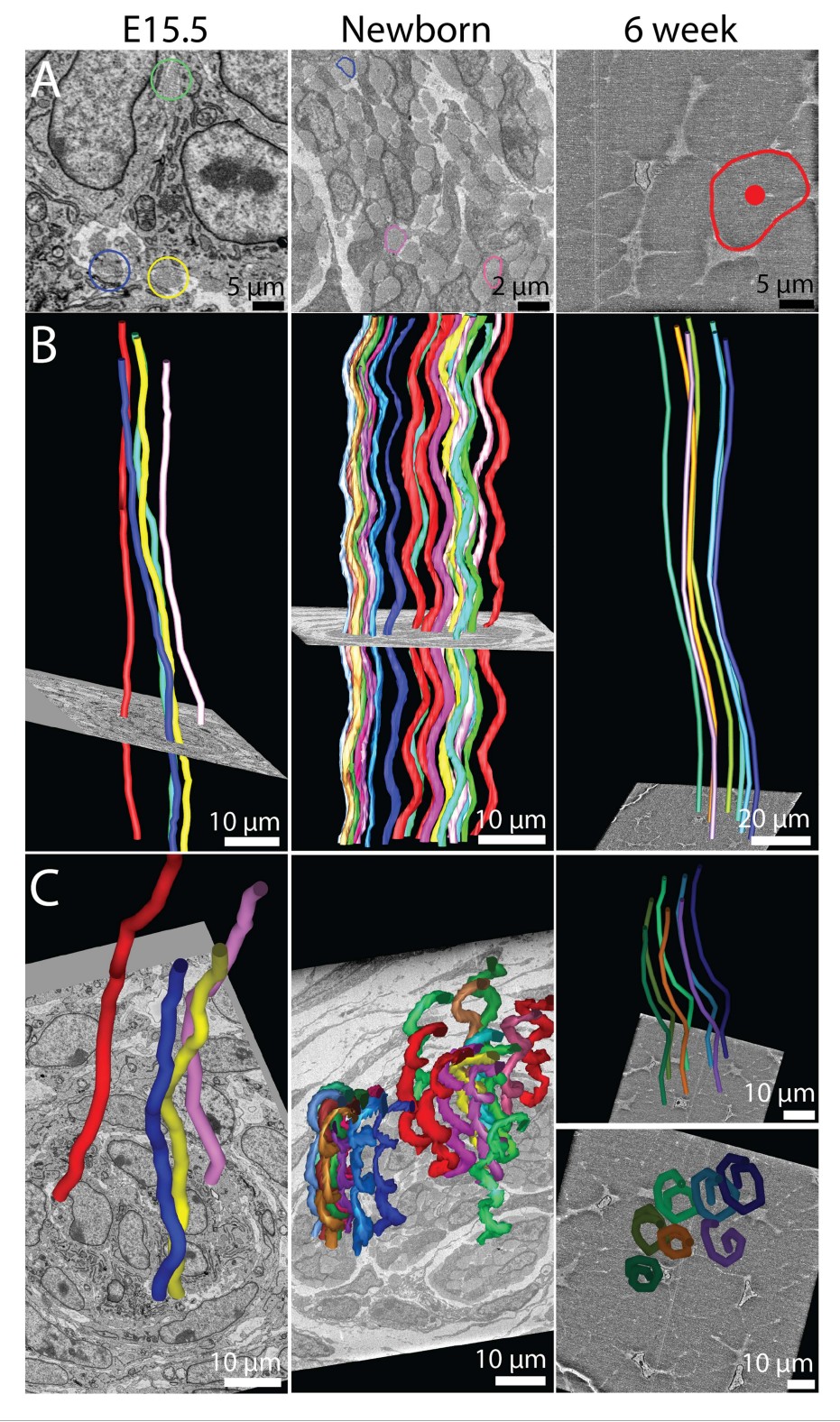

**Figure 7**. The 3D crimp structure of tendon. (**A**) Single SBF-SEM images of transverse sections at E15.5, newborn and 6 week old mouse-tail tendon. Fibril bundles are ringed in colour. At 6 weeks the centre of the bundle is marked with a red circle. Centre of bundle points were used to track the bundles in **B** and **C** (6 weeks, right hand panels). (**B**) Longitudinal views of bundle reconstructions. Colours correspond to bundles circled in **A**. The crimp is apparent

*Figure 7. continued on next page*

*Figure 7. Continued*

at E15.5, but it is not in register between different bundles. By birth the crimp is in register, all bundles have the same wavelength, in a helix throughout the tendon. This morphology is also present in 6 week tendon. (**C**) Oblique views of bundle reconstructions demonstrate that the crimp structure is a regular helix. A bird's-eye view of a fibril bundle reconstruction is shown for 6 weeks, demonstrating a spiral structure. Reconstructed bundles are tracked at E15.5 and newborn using the outline of the bundle. At 6 weeks the large size of the bundle made appreciation of the crimp structure difficult. Therefore the centre of the bundle was used as the marker point for reconstruction (marked in **A**, right hand panel).

The following source data and figure supplements are available for figure 7:

**Source data 1**. Crimp structure data.

**Figure supplement 1**. Crimp wavelength.

**Figure supplement 2**. Fibril length calculations.

---

An alternative mechanism of fibril growth is the surface-nucleation-and-propagation (SNP) model in which collagen molecules accrete onto the growing tips of fibrils (*Holmes et al., 1992*, *1998*) and to the molecular reversal region in bipolar fibrils (*Trotter et al., 1998*, *2000*). The SNP model predicts that collagen fibrils have specific binding sites for collagen molecules, leading to growth in diameter and length. Thus, fibril growth is determined by local structure at the fibril surface and is an example of interface-controlled growth as opposed to diffusion-limited accretion onto the fibril surface. The occurrence of an abrupt limitation in fibril diameter at the growing tips of vertebrate collagen fibrils also suggests a structure-based mechanism for fibril growth (*Holmes et al., 1998*). Evidence from collagen fibril diameter measurements in vivo also supports a structure-based assembly mechanism. If collagen molecules could assemble onto non-specific binding sites on the fibril in an uncoordinated manner, the axial mass profiles would be expected to show pronounced random fluctuations. In the case of diffusion-limited growth, collagen fibrils closer to the cell might be expected to have larger diameters because of the ready availability of collagen molecules emerging from the cell. However, as fibril diameters are independent of distance from the cell, it seems more likely that fibril growth (especially in postnatal tendon) is strongly influenced by the local structure of the fibril surface. Thus, the establishment of fibril number during stage 1 of development and subsequent growth in fibril diameter during stage 2 by the SNP mechanism (in the absence of overt shaft-to-shaft fibril fusion) would explain the observed increase in fibril diameter in stage 2 and the lateral growth of the tissue.

The ability of SBF-SEM to examine relatively large volumes of tissue at good resolution enabled us to assess cell shape with precision. As shown in *Figure 4*, cells at E15.5 and newborn had convoluted profiles, were slender cylinders with overlapping ends, and were ~50–60 µm in length. In contrast, cells in 6 weeks postnatal had prominent lateral cellular extensions that reached deep into the ECM, were short (~25 µm) and stacked in rows on their blunt ends. Importantly, cells at 6 weeks postnatal tendon remained in contact with each other via their cellular extensions regardless of the distance between the cell bodies where the nucleus was located. The number of collagen bundles at the cell periphery (and therefore the number of cell–cell contacts) was stable between newborn and 6 weeks postnatal. The data show that the main cell bodies move laterally apart because of the expansion of the ECM but maintain their longitudinal relationship, thereby generating the tramline organisation seen in mature tendon.

An unexpected result was visualisation of a spiral 3D organisation of collagen bundles to form the tendon crimp. In relaxed tendons, the bundles of collagen fibrils are buckled into a wavy pattern, called crimp, which is readily visible using plane polarized light (*Kapacee et al., 2008*). Unbuckling of the crimp during longitudinal loading functions as a natural shock-absorber on initial loading as well as being important in elastic recoil (reviewed by [*Benjamin et al., 2008*]). Crimp first appears during embryonic tendon development (*Shah et al., 1982*) but the mechanism of crimp formation remains unknown, largely because of the impracticalities of studying its formation in vivo. In our studies we were careful to preserve the tension in the tendon, avoiding major changes in crimp structure (*Shah et al., 1977*), by fixing whole tails without disturbing the axial attachment to the skeleton. Crimp 3D

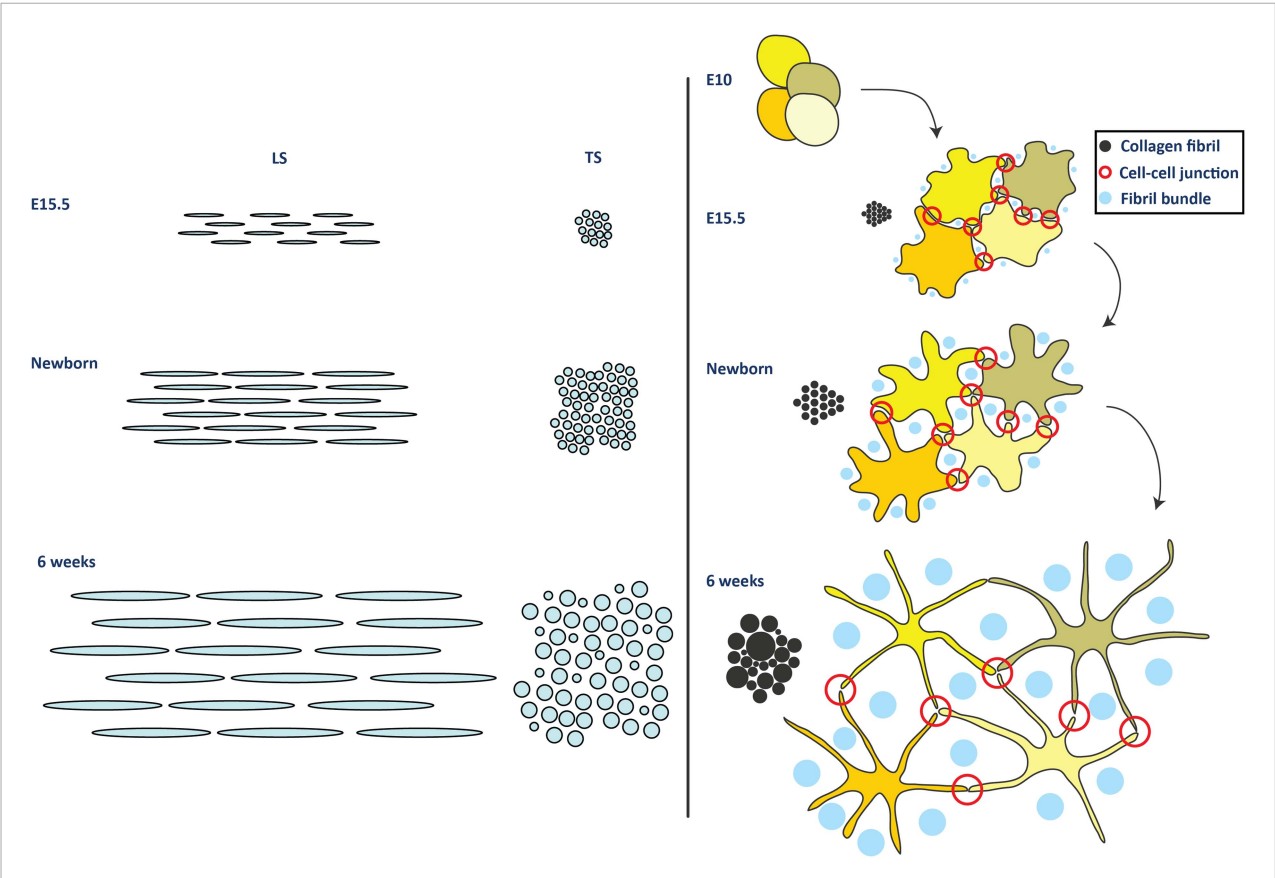

**Figure 8**. Model for development of tendon ECM. Left side panel. Schematic diagram to represent a model of fibril nucleation and growth during tendon development. Fibrillar arrays representing bundles are shown in longitudinal and transverse section (labelled LS and TS, respectively) at the 3 time points (E15.5, newborn and 6 week). The embryonic growth stage involves fibril nucleation to increase fibril numbers along with fibril growth to increase both fibril diameter and length. The numbers of fibrils in the transverse section of the bundle is increased not only by fibril nucleation but also by axial growth of the fibrils into neighbouring regions of the bundle. The postnatal growth in stage 2 involves only fibril growth, with a major increase in fibril diameter and length, with no increase in fibril number. To maintain the observed invariant number of fibrils in the bundle transverse section, the increase in fibril length must match the local increase in tissue length. Right side panel. Schematic to represent cell and fibril growth during tendon development. Early embryonic tendon tissue (E10) is composed of rounded cells with little ECM. Collagen fibrils begin to appear in small bundles in poorly defined channels formed by cell–cell adhesions by E15.5. These channels become more developed with expansion of the ECM and are well defined in newborn tendon tissue. Postnatal expansion of the ECM results in large fibril bundles in these channels that were established during embryonic development.

structure has been debated over a prolonged period. We found no kinks as previously described (*Raspanti et al., 2005*) but instead observed a left-handed spiral running the length of our 3D reconstructions. A spring/spiral nature of the tendon crimp has been predicted in a mathematical model (*Grytz and Meschke, 2009*). It is important to note that fibril bundles are continuous over the distances we have measured here, forming parallel spirals in axial register to provide uniform biomechanical properties when loaded, and there is a close relationship between the cell and the fibril bundle. The close association of cells with bundles suggests that the cell shape strongly influences the configuration of the crimp structure. This is consistent with studies suggesting that cell force generated by the actin-myosin system can create a crimp structure (*Herchenhan et al., 2012*). As illustrated in *Figure 7*, we showed that the crimp wavelength lengthens from ~14 µm at E15.5 to ~99 µm at 6 weeks postnatal. The length increase of the mouse-tail from newborn to 6 weeks by a factor of ~6.2× closely matches the increase in crimp wavelength of ~7.1×. This suggests a simple crimp elongation process in synchrony with bundle and tissue elongation.

In conclusion, we have shown that tendon development occurs in a two-stage process. In stage 1 the organisation of fibroblasts and the cellular channels for fibril deposition and growth are

established, and cell-mediated control of collagen fibril deposition occurs. This sets the developmental template for structure-based matrix expansion of the existing matrix in stage 2, which drives the increase in the size of the matrix that underpins tissue growth. Important features include the cell-regulated number of cell processes that define the boundaries of collagen fibril bundles and help to maintain the tramline organization of cells in the mature tissue. These mechanisms could be a paradigm for the growth of other matrix-rich tissues that are abundant in collagen fibrils.

## Materials and methods

### Mouse sample preparation

C57/Black6 mice were sacrificed using a UK Home Office approved S1 Schedule method. Male mice were sacrificed at embryonic day 15.5, birth (newborn) and at 6 weeks postnatal for SBF-SEM and TEM. Whole tendons were fixed in situ in the tail, preserving the native state of the tail tendon, which at E15.5 and at birth was small enough to allow penetration of the fixative. The skin from 6 week mouse-tail was removed, before fixation of the entire underlying structures to allow penetration of the fixative. Tendon from the tail axial mid-point was examined with TEM and SBF-SEM to allow comparison between different ages.

### SBF-SEM

Samples were prepared as described (*Starborg et al., 2013*). In brief, mouse-tails were fixed in situ by using 2% (wt/vol) glutaraldehyde (Agar Scientific, UK) in 0.1 M cacodylate buffer (pH 7.2), en-bloc stained in 1% (wt/vol) osmium tetroxide, 1.5% (wt/vol) potassium ferrocyanide in 0.1 M cacodylate buffer, followed by 1% (wt/vol) tannic acid in 0.1 M cacodylate buffer (pH 7.2). After washing with 0.1 M cacodylate buffer (pH 7.2), more osmium was added by staining in 1% (wt/vol) osmium tetroxide. The final staining step involved soaking in 1% (wt/vol) uranyl acetate. Samples were dehydrated in ethanol and infiltrated in Araldite resin (CY212; Agar Scientific).

A minimum of three samples per time point were imaged using SBF-SEM. In this experiment and other investigations in our laboratory we have imaged >100 mouse and chick tendons using SBF-SEM (12 were used for SBF-SEM in this study) (*Kalson et al., 2013*). Resin-embedded samples were sectioned using a Gatan 3View microtome within an FEI Quanta 250 scanning electron microscope as described (*Starborg et al., 2013*). For E15.5 and newborn mouse-tail samples, a 41 μm × 41 μm field of view was chosen and imaged by using a 4096 × 4096 scan, which gave an approximate pixel size of 10 nm. The section thickness was set to 100 nm in the Z (cutting) direction. For 6 week-old mouse-tail samples the field of view was increased to accommodate the larger cells (200 μm × 200 μm). Typically, Z volumes datasets comprised 1000 images (100 μm z depth). The IMOD suite of image analysis software was used to build image stacks, reduce imaging noise, and generate 3D reconstructions (*Kremer et al., 1996*). Cell volume, cell surface area, cell length and crimp wavelength were calculated using functions in the IMOD image analysis suite. 30 measurements (10 from each sample) of 3D cell length, 3D cell volume and 3D cell surface area were made for each time point. Cell length measurement was made along the tendon long axis as the distance between the tip and the tail of the cell. Measurement of crimp wavelength and helix radius was performed from reconstructions in IMOD. 50 measurements of crimp wavelength and helix radius were made at each time point (20 measurements from sample 1, 15 from samples 2 and 3).

Cell number measurement was made on three separate SBF-SEM samples for each time point (E15.5, newborn, 6 weeks). The volume of the tendon tissue in each SBF-SEM dataset was calculated and all the cells in the volume were reconstructed using IMOD. Each cell nucleus contained within the reconstruction was identified and counted. Cells per 1000 μm$^3$ of tissue were calculated to allow comparison between samples.

The longitudinal distance between nuclei in cell stacks in newborn and 6 week tendon was measured in three separate SBF-SEM datasets. 20 measurements were made on each sample as the distance between the mid-point (in the longitudinal plane) of each cell nucleus.

Cells per μm$^2$ of transverse tissue area and the transverse distance between cell nuclei in adjacent cell stacks in the transverse tissue plane were calculated from three separate SBF-SEM datasets at each time point (E15.5, newborn and 6 week for cells per μm$^2$, newborn and 6 weeks for transverse distance between nuclei). For each of these analyses 20 measurements per sample were made, each at 2.5 μm intervals through the tissue volume (along the longitudinal tendon axis). For cells per μm$^2$ all

cells in a single transverse image were counted, and the area of the tendon calculated. For distance between cell nuclei in adjacent cell stacks the distance between cell nuclei in the transverse tissue plane in cell stacks at newborn and 6 weeks was measured in three separate samples in newborn and 6 week tendon. Measurements were made between the edges of adjacent nuclei.

Cell–cell contacts were investigated in newborn and 6 week tendon. Three separate SBF-SEM datasets were used to generate cell reconstructions. For each dataset 10 individual cells were reconstructed, together with each cell's neighbouring cells. The number of cell protrusions that formed fibril channels was counted for each cell (30 cells in total). The number of different cells contacted by channel-forming cell membrane protrusions were counted for 10 cells per sample (30 cells in total).

Estimation of average fibril length was made as previously described (*Starborg et al., 2013*). For E15.5, 1000 fibrils in four different fibril bundles (two bundles from separate SBF-SEM datasets) were tracked for 10 μm (10,000 μm total length tracked). For newborn, 1000 fibrils from two separate SBF-SEM datasets (in four different bundles) were tracked over 20 μm (20,000 μm total length tracked). For 6 weeks, 1000 fibrils from two separate SBF-SEM datasets (four different bundles) were tracked over 10 μm (10,000 μm total length tracked). Total length of fibrils tracked and the number of fibril tips were recorded and the equation

$$L_m = \frac{2 \times \Sigma(\text{total length of fibrils in a bundle})}{\text{number of fibril tips in a bundle}}.$$

was used to estimate average fibril length.

## Transmission electron microscopy

For TEM analysis a minimum of three samples per time point (E15.5, newborn and 6 week) were prepared as previously described (*Starborg et al., 2013*; *Canty et al., 2004*). Sections were examined using an FEI Tecnai 12 Twin transmission electron microscope (TEM). Images were captured using a 2 k × 2 k cooled CCD camera (F214A, Tietz Video and Image Processing Systems, Gauting, Germany). For each sample a minimum of three different sections were reviewed. An unbiased sampling of each section was performed. Three different magnifications were used: 2100× for FAF and fibril bundle number per nucleus (giving a large-area survey), and 11,000× for fibril diameter, fibril area and fibrils per μm². The sampling procedure generated 20 images of each section at 2100×, 40 views per section at 6800× and 60 views per section at 11,000×. All measurements were made using ImageJ software (NIH freeware, http://rsb.info.nih.gov/nih-image). For fibril diameter measurement 500 fibrils were measured in total from three separate samples per time point (200 from sample 1, 150 each from sample 2 and 3). Fibril area was calculated from fibril diameter assuming circularity (1/4 × π × diameter²). Magnification calibration was performed for each magnification using a diffraction-grating replica grid (2160 lines/mm, Agar Scientific, Stansted, UK). 20 measurements of FAF were made for each time point (7 from sample 1, 7 from sample 2 and 6 from sample 3 [*Starborg et al., 2013*]). 30 measurements were made for fibrils per bundle, bundles per nucleus and fibrils per μm² at each time point (10 measurements from each sample) on TEM images of transverse sections of tendon. This gave a measurement of fibril bundles per nucleus in transverse section. Fibril bundles are continuous along the length of the tendon, so there are significantly fewer bundles than tendon cells when the tendon is considered as a whole composite unit containing cells and ECM.

## Tail length measurement

Measurements of tail length were made on four mice at E15.5, two newborn, and two 6 week mice (all male) using photographs of whole mice. Measurements were made using Image J.

## Cell isolation from tendon

C57/Black6 mice were sacrificed using a UK Home Office approved S1 Schedule method and tail tendons immediately dissected and the cells released by placing the tendon in trypsin (37,000 U) and bacterial collagenase (522 U) in DMEM at 37℃ for 2 hr. Cells were passed through a 70 μm cell strainer (BD Biosciences, UK), collected by centrifugation (240×g for 5 min) and washed 3 times in PBS. Cells were re-suspended in DMEM4 with 100 U/ml penicillin, 100 μg/ml streptomycin, 2 mM L-glutamine and 10% FCS. Cells were not passaged before examination by light microscopy. Three separate tendon cell isolations were performed for each time point.

## Light microscopy imaging of extracted tendon cells

Cells on coverslips were rinsed 3 times with PBS containing 0.9 mM $Ca^{2+}$ and 0.49 mM $Mg^{2+}$ (Sigma D8662) and fixed with 1% paraformaldehyde in 0.1 M HEPES (pH 7.4) for 15 min at room temperature. After being permeabilised cells were blocked with 1% BSA in PBS at room temperature for 30 min. FITC labelled phalloidin (Sigma) was added and incubated for 1 hr in the dark. Cells were washed, then left to air dry before mounting with vector shield containing DAPI and left to set at 4℃. Samples were examined with a Leica light microscope. Cell area was measured using ImageJ. 10 cells were measured from each isolate (n = 30 per time point).

## Immunofluorescence Cx32

Cryosections of mouse-tail tendon (10 μm) were fixed in 100% acetone at 20℃ for 10 min and blocked at 4℃ overnight with 5% normal goat serum in PBST (PBS supplemented with 0.1% Triton X-100). Sections were incubated with primary antibody (1:250) diluted in 1% bovine serum albumin in PBS for 1 hr, washed 3 times for 5 min each with PBST, and incubated with goat anti-rabbit–Cy3 (1:1000) for 1 hr. Tissue was washed 3 times for 5 min each with PBST and mounted with Vectashield mounting medium containing DAPI (4,6-diamidino- 2-phenylindole).

## Immunofluorescence Cx43

Cryosections of mouse-tail tendons (10 μm) were fixed in 2% PFA and blocked for 1 hr at 4℃ with 3% BSA in PBST (PBS supplemented with 0.1% Triton X-100). Sections were incubated with primary antibody (1:500) diluted in blocking buffer, overnight at 4℃ then washed 3 times for 5 min each with PBST, and incubated with goat anti-rabbit–Cy3 (1:1000) for 1 hr. Tissue was washed 3 times for 5 min each with PBST and mounted with Vectashield mounting medium containing DAPI.

Three separate tendon samples (three slides per sample) were stained for connexin 32 and 43. Images were collected on an Olympus BX51 upright microscope using a 20×/0.50 Plan Fln objective and captured using a Coolsnap ES camera using *MetaVue* Software (Molecular Devices). Images were then processed and analysed using ImageJ.

## Statistics

Data are presented as mean ± SEM. For all statistical tests type I error was set to 0.05 and p values less than 0.05 considered to be significant. Three groups were compared for all tests, so the one-way ANOVA was used with a Tukey's post-test. Tests were performed using SPSS version 20. A summary of raw data is presented in *Supplementary file 1*.

## Acknowledgements

The Wellcome Trust provided generous support to KEK to fund this work. The authors thank the staff in the EM facility in the Faculty of Life Sciences for their assistance, and the Wellcome Trust for equipment grant support to the EM facility.

## Additional information

### Funding

| Funder | Author |
| --- | --- |
| Wellcome Trust | Karl E Kadler |

The funder had no role in study design, data collection and interpretation, or the decision to submit the work for publication.

### Author contributions

NSK, DFH, Conception and design, Acquisition of data, Analysis and interpretation of data, Drafting or revising the article; YL, TS, Acquisition of data, Analysis and interpretation of data; SHT, Acquisition of data, Analysis and interpretation of data, Drafting or revising the article; KEK, Conception and design, Analysis and interpretation of data, Drafting or revising the article

### Author ORCIDs

Nicholas S Kalson, http://orcid.org/0000-0001-8394-3060

## Ethics

Animal experimentation: The care and use of all mice in this study was carried out in accordance with UK Home Office regulations, UK Animals (Scientific Procedures) Act of 1986 under the UK Home Office licence (PPL 40/3485). No experimental procedures were performed on live animals. All animals were sacrificed by Schedule 1 cervical dislocation by trained personnel, and all efforts were made to minimize suffering.

## Additional files

### Supplementary file

- Supplementary file 1. This file contains all the source data reported in the manuscript.

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
