## [Decision Letter]

Thank you for sending your work entitled “A structure-based expansion mechanism of fibrous tissue growth” for consideration at *eLife*. Your article has been favorably evaluated by Fiona Watt (Senior editor), Robb Krumlauf (Reviewing editor), and two reviewers.

The Reviewing editor and the reviewers discussed their comments before we reached this decision, and the Reviewing editor has assembled the following comments to help you prepare a revised submission.

It is the collective opinion of the reviewers that this work provides a 3D framework that supports and extends a large body of previous observations, over several decades, using classical electron microscopy and other techniques. In addition, the approach described provides important new structural insights into the old problem of how cells determine the way in which the extracellular matrix is assembled in order to form a functional tissue. This data provides novel mechanistic insight into how tissue growth can be mediated. Hence, this is exciting work that presents a new framework in which to think about tissue growth. However, before it can be recommended for publication in *eLife* there are two major concerns that need to be addressed.

The major issues are:

1) The authors need to demonstrate more evidence for their structure based expansion mechanism. If the authors can show that postnatal growth is primarily caused by the increase in fibril diameter size instead of any changes in cell number (or cell shape), this would further support their model. Hence it is important to address whether they also observe a change in cell number as this is another means of mediating growth. This analysis would be important specifically at the postnatal stages when the fibril bundle numbers per nucleus are constant, and could be addressed using the 3D reconstructions of the cells that they have already generated. Interestingly, the authors report that cell length decreases from 60 to 27 microns from embryonic to 6 week old stages, and it appears as though there are more cells that are shortened or more compacted along the longitudinal axis in Figure 3 (although their total volume remains the same). It would be interesting if the cells increase in number but also decrease their length, not contributing to the overall growth of the tissue. Because the cells are organized along tracks between the fibrils, does the number of cells in transverse cross-sectional area change differently or not at all compared with the numbers along the longitudinal axis? If there is an increase in cell number longitudinally, would this change the fibril bundle numbers per nucleus?

2) The authors state that cell-to-cell contacts are maintained during post-natal development (in the subsection headed “Increase in cell surface area during tendon development”), even though no quantitative data are presented to support this conclusion. Quantitative data that cell-cell contacts are maintained during postnatal development should be included in a revised version.

---

## [Author Response]

*1) The authors need to demonstrate more evidence for their structure based expansion mechanism. If the authors can show that postnatal growth is primarily caused by the increase in fibril diameter size instead of any changes in cell number (or cell shape), this would further support their model. Hence it is important to address whether they also observe a change in cell number as this is another means of mediating growth. This analysis would be important specifically at the postnatal stages when the fibril bundle numbers per nucleus are constant, and could be addressed using the 3D reconstructions of the cells that they have already generated. Interestingly, the authors report that cell length decreases from 60 to 27 microns from embryonic to 6 week old stages, and it appears as though there are more cells that are shortened or more compacted along the longitudinal axis in*
Figure 3
*(although their total volume remains the same). It would be interesting if the cells increase in number but also decrease their length, not contributing to the overall growth of the tissue. Because the cells are organized along tracks between the fibrils, does the number of cells in transverse cross-sectional area change differently or not at all compared with the numbers along the longitudinal axis*?

Thank you for the helpful comment. We have now undertaken new analyses relating to cell number during development. There data are presented in a new figure (Figure 5).

We first calculated the number of cells per unit volume of tissue for each developmental stage. We found a decrease in cells per µm^3^ between E15.5 and newborn tissue, and between newborn and 6 week tendon. The magnitude of this change was significant (almost 7 fold between newborn and 6 weeks; 0.91 to 0.14 cells per 1000 µm^3^).

The decrease in cell length (measured from tip to tail along the longitudinal axis of the cell) during development is an interesting observation, and following your suggestion we have explored this further. At E15.5 there are no distinct stacks of cells. Cell stacks are present at birth and at 6 weeks. The distance between nuclei of cells in stacks decreases from newborn to 6 week tissue, corresponding to a decrease in the longitudinal length of the cell. However, there is overlap of cells in newborn tissue, making the distance between nuclei less than the total length of the cell (40 vs. 55 µm). Cells in 6 week tissue are stacked end-to-end, with little overlap; their inter-nucleus distance corresponds closely to the length of the cell (28 vs. 27 µm). These data suggest that there is a relative increase in the number of cells in longitudinal axis in cell channels during tendon growth comparing newborn with 6 week tendon.

We have also analysed the changes in cell number in transverse section during development. The horizontal distance between cell nuclei in (measured in single transverse sections) is significantly increased from newborn to 6 week tendon tissue (from 6 to 32 µm). This corresponds to a decrease in cells per µm^2^ in transverse sections of 20.5±0.8 to 1.9±0.2 cells per 1000 µm^2^. This change is due primarily to the increase in size of the channels containing fibrils between the stacks of cells.

Taken together, these data suggest that although there is an increase in the number of cells longitudinally (in cell stacks) between newborn and 6 weeks, the major driver of tissue growth is an increase in the ECM in channels between cell stacks.

*If there is an increase in cell number longitudinally, would this change the fibril bundle numbers per nucleus*?

Thank you for the comment. We measured the number of fibril bundles per nucleus from transverse images. Therefore an increase in the longitudinal number of cells does not change this measurement. We now have clarified the method used to make this measurement in the Methods section.

These new data are described in the Results section, under the subtitle “Reduction in cell number per unit of tissue volume during development”.

*2) The authors state that cell-to-cell contacts are maintained during post-natal development (in the subsection headed “Increase in cell surface area during tendon development”), even though no quantitative data are presented to support this conclusion. Quantitative data that cell-cell contacts are maintained during postnatal development should be included in a revised version*.

Thank you for the helpful comment. We agree that this is a critical point and we have undertaken new experimental work and new analysis of 3D SBF-SEM data to investigate this question. These data are presented in a new figure (Figure 6 and Figure 6—figure supplement 1).

First we modeled cell-cell contacts at each developmental stage. At E15.5 no distinct cell membrane channels were formed via membrane protrusions contacting adjacent cells; instead groups of narrow-diameter fibrils were seen between adjacent cell membranes or enclosed in cell membrane invaginations (as shown in Figure 1 and in Figure 6). Newborn cells formed protrusions that contacted adjacent cells; these formed channels surrounding fibril bundles (Figure 6). Modeling of these cell-cell contact regions in 3D revealed close apposition of membranes along the longitudinal axis of the tissue, thereby defining the fibril channels. At 6 weeks a similar morphology of cell-cell contact was found, although the cell processes were much longer (Figure 6).

We have now quantitated these cell-cell contacts. In newborn tendon each tendon cell forms distinct channel -defining protrusions that contact, on average, 4.6 ± 0.1 adjacent cells. At 6 weeks each tendon cell contacts 4.5 ± 0.1 adjacent cells (p =>0.05, Figure 6—figure supplement 1). The number of protrusions per cell that contacted adjacent cells was also similar in newborn and 6 week tendon (8.2 ± 0.2 vs. 8.3 ± 0.2, Figure 6—figure supplement 1).

In addition we have undertaken immunofluorescence staining against connexins 43 and 32 in 6 week tendon tissue, and have now included data showing positive staining for connexins 43 and 32 (Figure 6). The extended longitudinal morphology of cell-cell contacts seen in both newborn and in 6 week tendon tissue, and the maintenance of similar numbers of cell-cell contacts and cell protrusions at birth and 6 weeks support suggest that the cell-cell contacts formed in embryogenesis form the basic pattern for the mature tendon tissue. Positive immunolocalisation of important gap junction connexins at 6 weeks supports this conclusion.

The increase in cell number along the longitudinal axis of the tendon in postnatal growth is compatible with this interpretation. New cells formed by mitosis during tissue growth can be added in cell stacks whilst at the same time maintaining the longitudinal channels containing fibril bundles, which are growing laterally due to the increase in fibril diameter.

This new data is presented in the subsection headed “Cell-cell contacts are maintained during development”, in the Results.

We have also included a new section in the Discussion regarding these new data, as follows:

“Although there is an increase in the number of cells longitudinally (in cell stacks) between newborn and 6 weeks, the fall in cells per unit volume of tissue due to expansion of the ECM suggest that the major driver of tissue growth is an increase in the ECM in channels between cell stacks. The extended longitudinal cell-cell contacts seen in both newborn and in 6 week tendon tissue, and the maintenance of similar numbers of cell-cell contacts and cell protrusions at birth and 6 weeks, suggests that the cell-cell contacts formed in embryogenesis form the basic pattern for the mature tendon tissue. Positive immunolocalisation of important gap junction components, connexin 32 and 43, at 6 weeks supports this conclusion.

The increase in cell number along the longitudinal axis of the tendon in postnatal growth is compatible with this interpretation. New cells formed by mitosis during tissue growth can be added in cell stacks whilst at the same time maintaining the longitudinal channels containing fibril bundles, which are growing laterally due to a dramatic increase in fibril diameter.”